# Physics-informed generative model for drug-like molecule conformers

## Abstract

Diffusion-based methods have been successfully applied to molecule conformer generation using implicit physical modeling. In contrast, conventional, rules-based approaches employ an explicit physical model such as a classical force field parameterization. In order to combine the advantages of both approaches, we present a diffusion-based, physics-informed denoising model (PIDM) for conformer generation that is constructed from molecule subgraph patterns borrowed from classical force fields. The result is a model that is resistant to overfitting and explainable. Using recent advances in denoising score matching, we naturally separate the task of training and generation while providing a smooth transition between deterministic and stochastic generative schemes that adapt to any number of denoising steps. We demonstrate conformer generation quality that outperforms the current state-of-the-art while employing a fraction of parameters.[1]

## 1 Introduction

Conformer generation is the process of identifying a valid and useful set of atomic coordinates for a given molecule. Because it plays a crucial role in structure-based drug-discovery [1], over four decades of effort has been invested in conventional rules-based approaches [2, 3, 4, 5, 6, 7, 8, 9]. Recent advances in generative techniques in deep learning, particularly in diffusion-based models of image generation [10, 11, 12, 13] and point-cloud generation [14, 15], suggest that generative techniques could lead to a renaissance in this field.

We describe in this work is a method of conformer generation using a physics-informed, denoising model (PIDM). By taking advantage of established methods employed in classical force fields, we have constructed a diffusion-based model that is explainable, transferable, and robust. Building upon recent theoretical advancements [16], we employ a flexible method of generation that smoothly adapts to either deterministic or stochastic modes and naturally scales to any number of denoising steps. This is the first time that a physics-informed approach has been applied in a denoising model of this type, the first that is explainable, and the first to feature deterministic generation. The result outperforms the current state-of-the-art while using ten times fewer steps. In addition, we demonstrate a proof-of-concept guided technique that permits targeted generation.

To build a useful tool, we need to generate molecule conformers that are physically meaningful. The naive approach is to directly train a generic, implicit model of physical viability. A drawback is that the number of possible drug-like compounds is too large (by some estimates, as large as $10^{33}$ [17]) to cover in a dataset. Another issue is that the number of viable conformations for each drug-like molecule is intractable in many cases [7]. Therefore, we are forced to limit ourselves to a suitable approach that can readily transfer to molecules and conformations outside a training set.

---

[1]Code is available at [TBA]

The first step is to define what constitutes a viable conformer. We use physics as a guide. To appreciate what constitutes a local minima in molecular energy, consider classic force field parameterizations, an established methodology [18]. Force fields are commonly formulated as the sum of "bonded" and "nonbonded" contributions [19, 20, 21]. The former include contributions associated with intramolecular bonds and the latter captures longer-distance interactions. The bonded contributions in a classical force field are conventionally divided into four terms: bonds, bends, proper torsions and improper torsions [19, 20, 21]. Each term is identified with a subgraph of a specific topology and one characteristic scalar parameter (Fig 1). These terms will become important in the construction of our model.

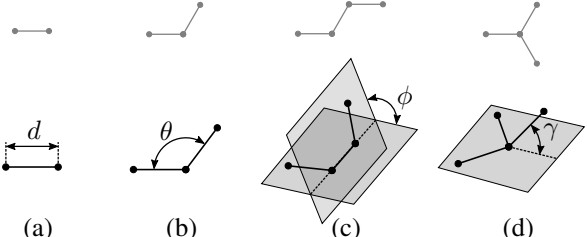

Figure 1: Force fields typically include bonded terms associated with (a) bond lengths, (b) bend angles, (c) proper torsions, and (d) improper torsions.

In the context of classical force fields, the set of viable molecule conformers are commonly defined in terms of fixed bond lengths, bond angles, and improper torsions. The constraint on proper torsions is cyclic in the angle $\phi$ such that energy is commonly parameterized as a function of $n\phi - \phi_0$, where $n$ is an integer and $\phi_0$ is a reference angle. The freedom to select one of $n$ torsion angles is one way a molecule can exhibit multiple conformations. We will refer to such freedom as *torsional space*.

Nonbonded contributions to energies depend on external conditions, such as whether the molecule is solvated or bound to a protein [22]. As such, in many applications, nonbonded energies are ignored or attenuated during the process of conformer generation, to avoid biases [23]. We employ this strategy in this work. It can be considered adequate for tools that perform their own search in torsional space, such as flexible ligand docking [24, 25, 26, 27, 28].

In addition to locating energy minima, a useful conformer generator must respect stable stereochemistry, specifically chirality [29] and cis-trans isomerism [30].

## 1.1 Related Works

Conventional approaches use a variety of schemes [4, 5, 6, 7, 8, 9, 24]. Balloon [5] is a conformer generator based on a multiobjective genetic algorithm. ETKDGv3 [9] is a knowledge-based generator provided by RDKit [31] based on distance geometry. The OMEGA toolkit [7, 32] employs a fragment library combined with rules-based sampling. Although widely adopted, conventional methods rely on hand-tuned algorithms and are typically limited in accuracy due to an overdependence on classical force fields parameterization.

Several strategies have been employed for learned models, such as energy gradients [33, 34, 35], Gibbs sampling [36], and conditional variational encoders [37, 38]. The drawback of these approaches is that the energy of disordered molecule systems is difficult to characterize directly due to singularities and large energy barriers. GeoMol [39] learns local structure and applies incremental construction. Since incremental construction is poorly suited to cycles, it fails to reproduce all but the simplest ring systems. GeoDiff [40] is a state-of-the-art stochastic diffusion model. It follows conventions most closely related to "denoising diffusion probabilistic models" (DDPM) [10], employs 793,858 weights and uses 5,000 steps for generation. GeoDiff attempts to model nonbonded distances which requires it to sample torsional space during training. This is not only undesirable (since torsional space is physically ambiguous) but is also likely the reason for a high level of computational complexity.

Others have proposed methods that generate novel molecules in 3D space [41, 42, 43, 44, 45, 46], which is a related but different task than reported here.

## 2 Model Design

Our goal is to construct a model that can generate acceptable conformers for any drug-like molecule when provided with just the atom composition, connectivity, and stereochemistry. Recent work [16] has demonstrated that the design of diffusion-based models for images can be generalized around the concept of denoising score matching [47]. We apply this approach to conformer generation by representing our model as a denoising function $D$ that provides an estimate of the true coordinates $\boldsymbol{x}$ of a molecule when provided with coordinates that have been smeared by a centered, uncorrelated Gaussian of width $\sigma$

$$\boldsymbol{x} \approx D(\mathcal{N}(\boldsymbol{x}; 0, \sigma^2 \boldsymbol{I}), \sigma; \boldsymbol{a}), \tag{1}$$

where $\boldsymbol{a}$ is a suitable embedding that represents the composition of the molecule (*i.e.* its atom types and connectivity).

The overall structure of the model is shown in Fig 2 and consists of two major components: a graph transformer network to build a useful atom embedding and a series of bonded subcomponents whose outputs are summed together for coordinate prediction.

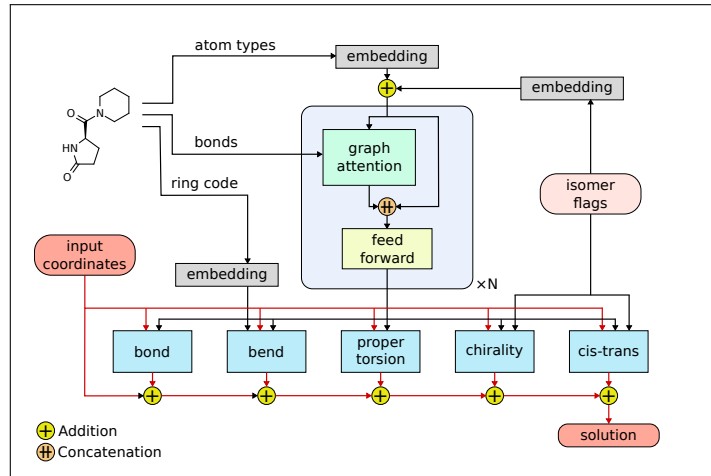

Figure 2: A schematic of the denoising model.

The purpose of the graph transformer network is to place the atoms of each molecule into a suitably descriptive embedding space that can be employed by the bonded components. It starts with a embedding in which atoms are distinguished by their element, formal charge, and hybridization. The latter is taken from the algorithm built into the RDKit cheminformatics library [31]. All hydrogens are treated as explicit. Atoms that are associated with explicit chirality or cis-trans isomerisms are flagged by the addition of a global vector reserved for this purpose.

The initial atom embedding is refined by multiple layers of a graph transformer network, based on GATv2 [48], and configured to use bonds as graph edges. No edge labeling (such as bond order) is employed, since connectivity along with atom identity is sufficient to describe relevant chemistry [49]. To preserve a form of atom self-identity, rather than use self edges (a poor physical analog), the input to the graph attention network is concatenated to the output.

Each of the bonded components have a similar structure in which a multilayer perceptron (MLP) is used to calculate a correction to atom positions as a displacement along a vector. The details, outlined in the supplemental materials, are summarized here. For the bond component, the vector is the difference $\boldsymbol{\delta}_{ij}$ of the two atom positions. The input to the MLP is the concatenation after a suitable normalization of $\{|\boldsymbol{\delta}_{ij}|, \sigma, \boldsymbol{a}_i, \boldsymbol{a}_j\}$, where $\boldsymbol{a}$ are the atom embeddings.

For the bend component, the vector $\boldsymbol{\delta}_{ik}$ is between the two outer atoms. The MLP is fed $\{|\boldsymbol{\delta}_{ik}|, \sigma, \boldsymbol{a}_i, \boldsymbol{a}_j, \boldsymbol{a}_k, \boldsymbol{c}_{ijk}\}$, where $\boldsymbol{c}_{ijk}$ is a embedding capturing the size of the rings (if any) to which the bend belongs. The latter is important because the GATv2 graph network, like all message passing networks, is incapable of detecting cycles [50].

For the proper torsion component, the vector $\boldsymbol{\delta}_{il}$ is between the two outer atoms. The MLP is fed $\{|\boldsymbol{\delta}_{il}|, \sigma, \boldsymbol{a}_i, \boldsymbol{a}_j, \boldsymbol{a}_k, \boldsymbol{a}_l, \sin\phi, \cos\phi\}$. The torsional angle $\phi$ is needed to account for torsional space.

The chirality component is constructed out of the improper torsions associated with each chiral atom. The vector is the normal of the plane defined by three of the four atoms. The MLP is fed $\{d_\perp, d_\parallel, \sigma, \boldsymbol{a}_i, \boldsymbol{a}_j, \boldsymbol{a}_k, \boldsymbol{a}_l\}$, where $d_\perp$ ($d_\parallel$) is the distance of the fourth atom out of (along) the plane. The cis-trans component is constructed out of the proper torsions associated with each constrained bond. The vector is the difference $\boldsymbol{\delta}_{ijkl}$ between the average positions of the outer and inner atoms. The MLP is fed $\{|\boldsymbol{\delta}_{ijkl}|, \chi_{ijkl}, \sigma, \boldsymbol{a}_i, \boldsymbol{a}_j, \boldsymbol{a}_k, \boldsymbol{a}_l\}$, where $\chi_{ijkl}$ is the requested cis-trans state.

For the models reported here, an atom embedding of dimension 50 is used throughout. Four graph transformer layers are employed. The MLP for each component uses two hidden layers. The result is a model with a total of 135,080 weights, with 63,480 reserved for the molecule graph and 71,240 in the geometry components. Experiments in increasing the atom embedding dimension or adding additional transformer layers produced only marginal improvement in loss.

## 3    Datasets

We are interested in high-quality conformers of drug-like molecules. Two synthetic, publicly available datasets fit this role: QMugs [51] and GEOM-drugs [52]. Each contain several hundred thousand drug-like molecules with conformers optimized (in vacuum) using the GFN2-xTB semiempirical quantum mechanical method [53, 54]. We discard about 2% of the GEOM-drugs molecules due to conformer inconsistency at the graph level. Otherwise, we accept all conformers in both data sets. Both are randomly divided into their own training (80%), validation (10%), and test (10%) subsets.

To measure molecule similarity, we use the Tanimoto measure applied to an ECFP6 fingerprint [55] folded to 1024 bits. At a threshold of 0.9 (0.8), we find 5.6% (6.4%) of the compounds in QMugs overlap with those in GEOM-drugs. Although the two datasets contain mostly different molecules, the underlying physics will be the same. To test dataset independence, we have chosen to train two versions of our model on the two corresponding training subsets, which we will label PIDM[Qmugs] and PIDM[GEOM-drugs]. Both models and others will be compared to a single benchmark dataset, whose preparation is described below.

For the benchmark dataset, we started from the QMugs test subset. To ensure independence and guard against data leakage, we filter each molecule against the contents of the QMugs training subset, the entire GEOM-drugs dataset, and internally using a Tanimoto threshold of 0.7. The annotated chirality and cis-trans isomerism are then validated against the public PubChem database [56]. We also queried PubChem for a copy of the first ten of their generated conformers [23] for comparison. Molecules that could not be validated or did not have a PubChem conformer were discarded. The final result is 15,763 fully annotated, independent molecules reserved for benchmarks.

## 4    Training

In this work, we follow the modular scheme for diffusion-based models as proposed by Karras et al. [16] in which training and generation are separate tasks that need not share the same noising schedule. For the purposes of training, we chose to evenly sample from the set $\{\sigma_1 \dots \sigma_N\}$:

$$\sigma_i = \begin{cases} \left(\sigma_{\max}^{1/\rho} + \frac{i-1}{N-1}\left(\sigma_{\min}^{1/\rho} - \sigma_{\max}^{1/\rho}\right)\right)^\rho & 1 \leq i < N \\ 0 & i = N \end{cases}. \tag{2}$$

The total loss $\mathcal{L}$ is calculated as the weighted sum of the contribution from each sample $\sigma_i$

$$\mathcal{L} = \sum_{i=1}^{N} \frac{1}{\sqrt{\sigma_i^2 + \epsilon^2}} \mathcal{L}(\sigma_i) \tag{3}$$

with

$$\mathcal{L}(\sigma) = \mathbb{E}_{\boldsymbol{x} \sim \text{data}, \boldsymbol{n} \sim \mathcal{N}(0, \sigma^2 \boldsymbol{I})} \|\boldsymbol{D}(\boldsymbol{x} + \boldsymbol{n}, \sigma) - \boldsymbol{x}\|_2^2. \tag{4}$$

Standard parameter values are $N = 100$, $\sigma_{\max} = 8\text{Å}$, $\sigma_{\min} = 10^{-5}\text{Å}$, $\epsilon = 10^{-5}\text{Å}$, and $\rho = 6$.

PIDM[Qmugs] is trained using a fixed schedule of 100 epochs, taking approximately 50 hours on a single RTX 3090 (Fig. 3a). PIDM[GEOM-drugs] is trained using a fixed schedule of 25 epochs, taking approximately 170 hours (Fig. 3b).

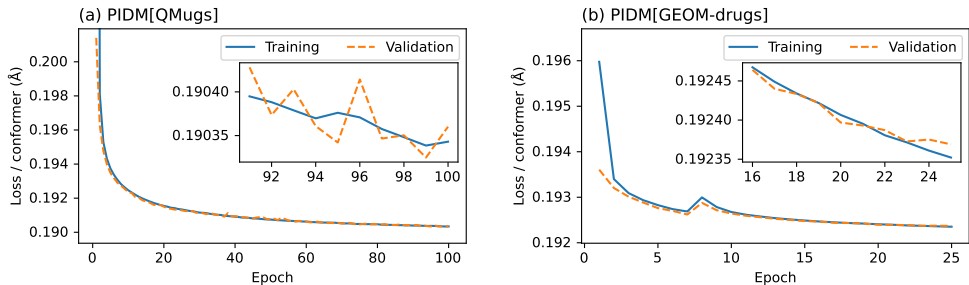

Figure 3: Loss per conformer as calculated during model training for (a) QMugs and (b) GEOM-drugs. Plotted are losses calculated for the training set and for an independent validation set of $1/8$ the size.

Building the model with explicit components has the advantage of explainability. Once provided with an example molecule, each of the bonded compounds can be probed for specific atoms, a process that was invaluable during development. An example is shown in Fig 4. More examples are included in the supplemental materials.

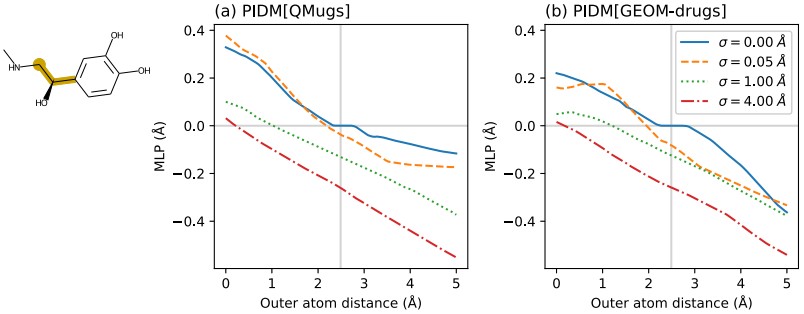

Figure 4: Example bend correction from (a) PIDM[QMugs] and (b) PIDM[GEOM-drugs] for an atom involved in the ethanol group of adrenaline. Corrections for various values of $\sigma$ are plotted. The vertical gray line is the expected atom distance, obtained from a separate GFN2-xTB optimization. As the distance deviates from expectations, the model applies larger corrections. As $\sigma$ approaches zero, the correction vanishes once the correct geometry is achieved. Despite being trained on independent data sets, the two models learn a similar behavior.

# 5  Generation

We adopt a score-based, probability-flow framework [13] in order to generate conformers from our denoising model. As is typical in this approach, we consider a multidimensional Wiener process applied to molecule coordinates $\boldsymbol{x}$ over a time interval $t \in [0, 1]$:

$$p_t(\boldsymbol{y}(t)|\boldsymbol{x}; \sigma(t)) = \mathcal{N}(\boldsymbol{y}(t); \boldsymbol{x}, \sigma(t)^2 \boldsymbol{I}), \qquad (5)$$

where $\boldsymbol{y}(t)$ are the resulting random coordinates and $\sigma(t)$ is a width schedule we are free to choose to suit our task, with the only requirement that $\lim_{t \to 0} \sigma(t) = 0$. For generation, we start by sampling from a random Gaussian distribution $\mathcal{N}(0, \sigma(1)^2 \boldsymbol{I})$ as an approximation for $\boldsymbol{y}(1)$ and solve for the corresponding reverse process (denoising) to obtain $\boldsymbol{y}(0)$ as a candidate solution for $\boldsymbol{x}$.

To construct a solution for the reverse process, we identify the marginal distribution $p(\boldsymbol{y}; \sigma)$ as

$$p(\boldsymbol{y}; \sigma) = \int p_t(\boldsymbol{y}|\boldsymbol{x}; \sigma) \, p(\boldsymbol{x}) \, d\boldsymbol{x}, \qquad (6)$$

where the $t$ dependence is implicit and $p(\boldsymbol{x})$ represents the marginal distribution of the training data. We can use $p(\boldsymbol{y}; \sigma)$ to express the time dependence of $\boldsymbol{y}$ as a *probability-flow* ODE [16]:

$$d\boldsymbol{y} = -\dot{\sigma}\sigma \, \nabla_y \log p(\boldsymbol{y}; \sigma) \, dt, \qquad (7)$$

where $\nabla_y \log p(\boldsymbol{y}; \sigma)$ is the *score function* and $\dot{\sigma}$ is the time derivative. In a score-based framework, there is a direct relationship between the score function and our denoising model $D$ [16, 47]:

$$\nabla_y \log p_t(\boldsymbol{y}; \sigma) \approx \frac{1}{\sigma^2} \left( D(\boldsymbol{y}, \sigma) - \boldsymbol{y} \right) . \tag{8}$$

This important result connects our denoising model to the conformer generation process.

In our implementation, we have selected a linear function $\sigma(t) = \alpha t$, where $\alpha$ is a scale parameter in units of Å. Applying this selection to Eq. 7 and 8 results in a simple form for the probability-flow ODE:

$$\frac{d\boldsymbol{y}}{dt} = \left( \boldsymbol{y} - D(\boldsymbol{y}; \alpha t) \right) / t . \tag{9}$$

Our conformer generation process is the numerical solution to this equation, calculated in steps of $t$ in reverse, and using as initial conditions $\boldsymbol{y}(1) \sim \mathcal{N}(0, \alpha^2 \boldsymbol{I})$.

Inspired by work elsewhere [16], we solve Eq. 9 using Heun's 2nd-order method, augmented by a form of backtracking (Algorithm 1). The backtracking provides an option to add additional noise to the generation process. We begin by dividing the interval $[0, 1]$ into a fixed partition $\{t_i\}$ over which we iterate in order to calculate a set of intermediate solutions $\{\boldsymbol{y}_i\}$. Instead of relying on solving on the partition $\{t_i\}$, we substitute modified values for the upper bound $t_i$ of each subinterval:

$$\tilde{t}_i = \beta t_i \quad \tilde{\boldsymbol{y}}_i \sim \mathcal{N}(\boldsymbol{y}_i; 0, \lambda^2 \alpha^2 t_i^2 (\beta^2 - 1)) , \tag{10}$$

where $\beta \geq 1$ and $\lambda \geq 0$ are fixed parameters. This has the effect of introducing Gaussian noise at each step of the solution. For $\lambda = 1$, the amount of added noise compensates for the change in subinterval size.

---

**Algorithm 1** Conformer generation.

---

1: **procedure** GENERATE($D(\boldsymbol{y}, \sigma), \{t_i\}, \alpha, \beta, \lambda$)
2: $\quad \boldsymbol{y} \leftarrow \mathcal{N}(0, \alpha^2 \boldsymbol{I})$ $\qquad\qquad\qquad\qquad\qquad\qquad\qquad$ ▷ Prepare random initial state
3: $\quad$ **for** $i \leftarrow 1$ to $|t|$ **do**
4: $\qquad \tilde{t} \leftarrow \beta t_i$ $\qquad\qquad\qquad\qquad\qquad\qquad\qquad\qquad$ ▷ Widen effective subinterval
5: $\qquad \tilde{\boldsymbol{y}} \leftarrow \mathcal{N}(\boldsymbol{y}; 0, \lambda^2 \alpha^2 t_i^2 (\beta^2 - 1) \boldsymbol{I})$ $\qquad\qquad\qquad\qquad$ ▷ Add noise
6: $\qquad \boldsymbol{d}_1 \leftarrow \left( \tilde{\boldsymbol{y}} - D(\tilde{\boldsymbol{y}}, \alpha \tilde{t}) \right) / \tilde{t}$ $\qquad\qquad\qquad\qquad\qquad$ ▷ Evaluate $d\boldsymbol{y}/dt$
7: $\qquad \boldsymbol{y} \leftarrow \tilde{\boldsymbol{y}} + \left( t_{i+1} - \tilde{t} \right) \boldsymbol{d}_1$ $\qquad\qquad\qquad\qquad\qquad\qquad$ ▷ Solve
8: $\qquad$ **if** $t_{i+1} > 0$ **then**
9: $\qquad\qquad \boldsymbol{d}_2 \leftarrow \left( \boldsymbol{y} - D(\boldsymbol{y}, \alpha t_{i+1}) \right) / t_{i+1}$ $\qquad\qquad$ ▷ Apply 2nd-order correction
10: $\qquad\qquad \boldsymbol{y} \leftarrow \tilde{\boldsymbol{y}}_i + \frac{1}{2} \left( t_{i+1} - \tilde{t} \right) (\boldsymbol{d}_1 + \boldsymbol{d}_2)$
11: $\qquad \boldsymbol{y} \leftarrow \boldsymbol{y} - \langle \boldsymbol{y} \rangle$ $\qquad\qquad\qquad\qquad\qquad\qquad$ ▷ Remove center of mass
12: $\quad$ **return** $\boldsymbol{y}$

---

For reasons of convenience, we remove an overall center-of-mass during each generation step. The correction is small and quality of output is not affected.

If we generate using $\lambda = 0$, no noise is added during the intermediate steps, resulting in deterministic generation. Combined with $\beta > 0$, the algorithm is equivalent to pretending that each intermediate value $\boldsymbol{y}_i$ belongs to a solution sampled from a larger value of $\sigma$. This has the effect of overcorrecting, which improves accuracy in our case.

Alternatively, if we generate using $\lambda > 0$ and $\beta > 0$, we inject noise during each step for stochastic generation. Both stochastic and deterministic approaches have been used for image generation, with impressive results [10, 11, 12, 13, 16, 57, 58, 59].

To apply our algorithm, we use the partition $\{t_1 \ldots t_N\}$ of a given size $N$ and final step size $t_\epsilon$:

$$t_i = \begin{cases} t_\epsilon^{i/(N-1)} & 1 \leq i < N \\ 0 & i = N \end{cases} . \tag{11}$$

The quality of generated output is reasonably stable for a large range of parameter values. The results reported here use $t_\epsilon = 0.0006$, $\alpha = 2.5$ Å, and $\beta = 5$. Quality improves marginally if the solution is

195 calculated using more steps at a proportional cost in processing time. To quantify this trade off, we
196 report on results for $N = 100, 200,$ and $500$. We also report results for both deterministic ($\lambda = 0$)
197 and stochastic ($\lambda = 1$) generation.

198 Shown in Fig 5 are random examples of generated conformers, using PIDM[QMugs], deterministic
199 generation, and 500 steps. More examples are available in the supplemental materials.

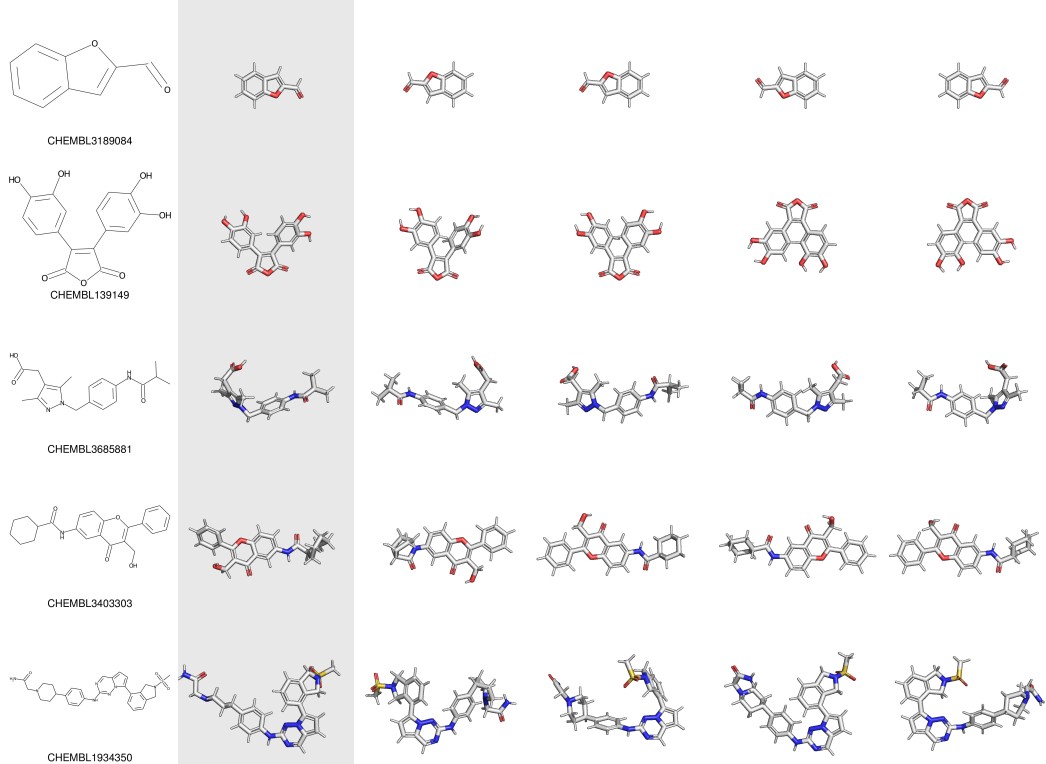

Figure 5: Example conformer output for molecules randomly selected from the benchmark dataset.
Shown in the second column from the left (grey background) is the first conformer from QMugs.
Shown on the right are unfiltered output from PIDM[QMugs] using deterministic generation and 500
steps. All molecule renderings are oriented by principal component.

## 6 Experiments

201 Our objective is to generate molecules with valid bonded geometry. To demonstrate accuracy, ten
202 random conformers were generated by PIDM[QMugs] and PIDM[GEOM-drugs] for each of the
203 molecules in the benchmark set. Each result was compared against the ground truth represented by
204 the corresponding conformers in QMugs. Overall errors in bond length $d$ and bend angle $\theta$ were
205 measured using the mean absolute deviation (MAD). Because proper torsions have multiple favored
206 values, the MAD for $\phi$ was limited to values generated with $\pm 30°$ of the angle found in the reference
207 conformer. Failures to reproduce the desired chirality or cis-trans isomerism were recorded as a
208 fraction of total occurrence of associated improper or proper torsion.

209 Results are shown in Table 1. Also shown are results obtained from other conformer generation
210 solutions. In all cases, ten conformers were requested, although some conformer solutions provide
211 less than the requested number under certain circumstances. The conformers provided by PubChem
212 are calculated under the PubChem3D scheme [23] based on the OMEGA toolkit [32]. ETKDGv3
213 has known deficiencies and is often followed by gradient optimization based on the MMFF94 force
214 field [6, 21]. We used the GeoMol checkpoint (GEOM-drugs) provided by the authors.

215 We tested the GeoDiff checkpoint trained on GEOM-drugs provided by the authors. This model
216 performs well on $d$ and $\theta$, but struggles with $\phi$. GeoDiff has no mechanism for enforcing chirality

Table 1: Performance data for PIDM[QMugs] and PIDM[GEOM-drugs] for 100, 200 and 500 steps, using both deterministic and stochastic schemes, and with optional guided generation ($\eta > 0$, see section 7). Also shown are other methods. All statistics are measured against a single, independent benchmark dataset. Best values in each category are highlighted.

| | | Mean absolute deviation | | | Inconsistency rate | |
|---|---|---|---|---|---|---|
| | | $d$ (Å) | $\theta$ (rad) | $\phi$ (rad) | chirality | cis-trans |
| PIDM[Qmugs] | | | | | | |
| Deterministic | 100 | 0.0042 | 0.015 | 0.036 | 0.031 | 0.033 |
| Deterministic | 200 | 0.0038 | 0.013 | 0.027 | 0.022 | 0.009 |
| Deterministic | 500 | 0.0036 | 0.012 | 0.023 | 0.013 | 0.002 |
| Deterministic | 500, $\eta = 0.5$ | 0.0035 | 0.012 | 0.025 | 0.028 | 0.001 |
| Deterministic | 500, $\eta = 1$ | 0.0036 | 0.012 | 0.026 | 0.057 | 0.001 |
| Stochastic | 100 | 0.0051 | 0.021 | 0.079 | 0.112 | 0.027 |
| Stochastic | 200 | 0.0047 | 0.019 | 0.069 | 0.081 | 0.013 |
| Stochastic | 500 | 0.0045 | 0.018 | 0.062 | 0.057 | 0.004 |
| PIDM[GEOM-drugs] | | | | | | |
| Deterministic | 100 | 0.0044 | 0.015 | 0.034 | 0.031 | 0.034 |
| Deterministic | 200 | 0.0040 | 0.013 | 0.027 | 0.023 | 0.015 |
| Deterministic | 500 | 0.0037 | 0.012 | 0.023 | 0.015 | 0.005 |
| Deterministic | 500, $\eta = 0.5$ | 0.0036 | 0.012 | 0.025 | 0.029 | 0.002 |
| Deterministic | 500, $\eta = 1$ | 0.0036 | 0.012 | 0.025 | 0.048 | 0.003 |
| Pubchem3D (OMEGA) | | 0.0075 | 0.020 | 0.020 | 0.020 | 0.014 |
| ETKDGv3 | | 0.0183 | 0.039 | 0.019 | 0.049 | 0.017 |
| ETKDGv3+MMFF94 | | 0.0081 | 0.017 | 0.021 | 0.049 | 0.017 |
| Balloon | | 0.0082 | 0.018 | 0.032 | 0.002 | 0.011 |
| GeoMol | | 0.0125 | 0.030 | 0.042 | 0.032 | 0.087 |
| GeoDiff | | 0.0051 | 0.017 | 0.170 | 0.500 | 0.263 |

nor cis-trans isomerism, an omission that appears to be an oversight rather than a limitation of the approach.

# 7 Guided Generation

Our model makes no attempt to predict distances between nonbonded atoms. This suggests that there are degrees of freedom available during generation that are being ignored. Addressing this deficiency could provide a valuable measure of control over the torsional space.

Before experimenting, we require a control. The torsional space sampled in the synthetic QMugs and GEOM-drugs data sets were established by their authors based on explicit yet arbitrary criteria, which makes them uninteresting. Instead, we can refer to experimental data. The authors of the OMEGA toolkit selected two small experimental sets for this purpose [32]: 480 molecules from the Cambridge Structural Database (CSD) and 197 ligands from the PDB. As shown in Table 2, RMSD performance of our model lags that obtained from OMEGA and RDKit+MMFF94.

Consider a modified, probability-flow ODE:

$$\frac{d\boldsymbol{y}}{dt} = (\boldsymbol{y} - D(\boldsymbol{y}; \alpha t) - F(\boldsymbol{y}))/t\,, \tag{12}$$

where $F(\boldsymbol{y})$ is introduced to guide generation in a desired fashion. Solving for Eq. 12 in place of Eq. 9 provides a mechanism for guided generation where $F(\boldsymbol{y})$ serves as a form of conditional score [60].

As a simple proof-of-concept, consider $F(\boldsymbol{y}) = \eta \sum (b^2 + \delta^2)^{-5}$, summed over all nonbonded atom pairs, where $\delta$ is the distance between atoms, $b = 0.7$Å, and $\eta$ is an overall magnitude. This term is analogous to a repulsive force similar to what is found in the van der Waals interaction. The result is a modest improvement in RMSD statistics on the CSD and PDB experimental data sets, as shown in Table 2. Statistics on bond parameters (Table 1) are largely unaffected, with the notable exception of an increase in chirality failure rates.

Table 2: RMSD statistics (Å, heavy atoms only) on conformers generated by PIDM (deterministic with 500 steps) and various other methods compared to experimental data from the CSD and PDB. Mean and median are calculated on the closest conformer out of $N$ generated.

| Model | | N | CSD | | PDB | |
|---|---|---|---|---|---|---|
| | | | Mean | Median | Mean | Median |
| PIDM[QMugs] | Undirected | 1000 | 0.74 | 0.84 | 0.90 | 0.98 |
| PIDM[QMugs] | $\eta = 0.5$ | 1000 | 0.53 | 0.58 | 0.70 | 0.75 |
| PIDM[QMugs] | $\eta = 1$ | 1000 | 0.51 | 0.55 | 0.67 | 0.73 |
| PIDM[GEOM-drugs] | Undirected | 1000 | 0.78 | 0.91 | 0.96 | 1.10 |
| PIDM[GEOM-drugs] | $\eta = 0.5$ | 1000 | 0.54 | 0.59 | 0.81 | 0.81 |
| PIDM[GEOM-drugs] | $\eta = 1$ | 1000 | 0.51 | 0.55 | 0.69 | 0.76 |
| OMEGA[a] | | — | 0.51 | 0.44 | 0.67 | 0.51 |
| RDKit+MMFF94 | | 1000 | 0.43 | 0.48 | 0.53 | 0.64 |

[a] Published statistics [32]

# 8 Discussion

The PIDM[QMugs] and PIDM[GEOM-drugs] models, despite being trained on datasets with little overlap, perform similarly on all benchmarks. Probes indicate that both models have learned similar solutions. We contend that this level of robustness could only have been achieved by capturing the important, underlying physics. Unlike conventional approaches to conformer generation, which either rely on curated template libraries or manually-tabulated force field parameterizations, the physics in our model is entirely learned.

PIDM is explainable. The advantage this provides cannot be overstated. The ability to probe the internal structure during development permitted a level of experimentation that would not have been possible in a black-box implementation.

For generation, we numerically solve for the probability-flow ODE, using a form of oversampling. By separating the training and generation tasks, we were free to explore multiple approaches without retraining. Using deterministic generation, we were able to produce conformers of reasonable quality in as little as 100 steps. This is in stark contrast with GeoDiff [40], the current state-of-the-art diffusion model, which employs stochastic generation in 5,000 steps.

For reproducing bonded parameters, PIDM with 500 steps has comparable performance on average to conventional methods (such as OMEGA and ETKDGv3). It outperforms GeoDiff with 1/6 the number of weights.

## 8.1 Limitations

Conformer generation for molecules with chemical groups or atom types outside the training set may perform poorly or fail. Molecules with certain challenging topologies, such as a central ring with several large branches, may perform poorly (an example is given in the supplemental materials). Conformer quality is expected to degrade as molecules grow in size beyond $\sim$200 heavy atoms.

The atom embedding used in our model was generated using GATv2 [48]. We suspect that using a type of graph network that can capture the same quality of atom-type information while also recognizing cycles would improve performance.

# 9 Conclusion

Presented is PIDM, a diffusion-based model inspired by the bonded components of classical force fields. Parameters were trained on high-quality conformers from the QMugs and GEOM-drugs data sets. Learning appears robust, transferable, and explainable. Both deterministic and stochastic generation schemes are demonstrated. Average performance on the reproduction of bonded parameters is comparable to conventional conformer generation tools. A simple example of guided generation is successful at improving torsional sampling when compared to experimental data.

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
