# OpenReview forum: "Physics-informed generative model for drug-like molecule conformers"
_NeurIPS.cc/2023/Conference — Submitted to NeurIPS 2023_

### Official Review · Reviewer_J8rt · 2023-07-04

**Soundness:** 2 fair
**Presentation:** 2 fair
**Contribution:** 2 fair
**Rating:** 3
**Confidence:** 4

**Summary:**

This paper proposes PIDM, a novel generative model for generating 3D molecular conformations from 2D molecular graphs. The proposed method uses an ODE based diffusion model to generate 3D molecular conformations, and designs a denoising network with multiple different modules to capture various of physical information from molecules. The proposed method achieves better performance than two baseline methods, GeoMol and GeoDiff, in experiments.

**Strengths:**

- This work proposes a series of novel strategy to capture geometric information from different geometric inputs (e.g., bonds, bends, torsions) by different modules in a model. The proposed model architecture can be useful for a variety of molecular machine learning tasks.
- Experimental results show that the proposed method achieves promising performance when compared with two strong baseline methods, GeoMol and GeoDiff, in benchmark datasets.

**Weaknesses:**

Major:
- A major weakness of this paper lies in that many necessary details of the proposed method is not presented or not sufficiently described as detailed below.
(1) The proper torsions and improper torsions introduced in line 40-41 are not formally defined and it is not easy to discriminate them by Figure 1. Also, it is not clear why the model has a module for proper torsions but no module for improper torsions.
(2) It is not easy to understand why "The constraint on proper torsions is cyclic in the angle $\phi$ such that energy is commonly parameterized as a function of $n\phi-\phi_0$" (line 46). Does it mean that proper torsions are periodic and $\phi$ is the periodicity? How this constraint is incorporated in the designed model?
(3) The details of several modules are not described in Section 2. The details of the computation process in graph attention module is not described. The cis-trans module in Figure 2 is not introduced. Also, it is not clear what is the output ("solution" in Figure 2) of the model, is it the atom embedding, or the denoised atom coordinate? How the outputs are computed if they are not atom embeddings?
- For experiments, it would make results stronger if authors can do comprehensive ablation studies about the proposed novel modules in the model as they are the major novelty contributions. Also, it is recommended to compare with an important baseline method, Torsional Diffusion [1].

Minor:
- To make a better organization, it would be better to move Section 3 Datasets to the place just before Section 6 Experiments as they are more closely related.
- It is hard to understand what Figure 4 tries to show. What do the y-axis values mean? And how these curves are related to explainability? Authors may make more clarification, and are recommended to move these results to experiment part to maintain a better organization.

[1] Torsional Diffusion for Molecular Conformer Generation. NeurIPS 2022.

**Questions:**

No additional questions.

**Limitations:**

Yes.

---

> ### Author Rebuttal · Authors · 2023-08-08
>
> Thank you for reviewing our submission and providing valuable feedback.
>
> Since you have provided no explicit questions, we will attempt to address your list of weaknesses.
>
>  * The proper torsions and improper torsions introduced in line 40-41 are not formally defined and it is not easy to discriminate them by Figure 1
>
> The definitions of these terms are standard and available from the references [19-21]. We apologize if the figures are not clear. Explicit equations are provided in the supplemental materials.
>
>  * Also, it is not clear why the model has a module for proper torsions but no module for improper torsions
>
> We explain this in the supplemental materials:
> > “No generic improper torsion component is provided because improper torsions are already constrained by bond lengths and bond angles. Force field parameterizations typically include a select set of improper torsions primarily as a means to enforce planarity in conjugated systems. This is needed to characterize forces which are calculated as first order derivatives of energy with respect to atom position. Such functionality is not needed here.”
>
>  * Does it mean that proper torsions are periodic and ɸ is the periodicity?
>
> We can appreciate the confusion. This sentence should probably have been stated as the following: “The constraint on proper torsions is cyclic in the angle ɸ such that energy is commonly parameterized as a trigonometric function of nɸ + ɸ0”
>
>  * How this constraint is incorporated in the designed model?
>
> By using cosɸ and sinɸ as inputs to the MLP, we ensure a periodic function.
>
>  * The details of several modules are not described in Section 2
>
> All properties of these modules are explained in sufficient detail to reproduce their general form and understand their construction. Equations are provided in the supplementary materials. The details can be cross-referenced with the implementation, which is provided.
>
>  * The details of the computation process in graph attention module is not described.
>
> The GATv2 model is a standard technique and a suitable reference is provided.
>
>  * The cis-trans module in Figure 2 is not introduced.
>
> Incorrect. It is referred to by name in lines 113-115.
>
>  * Also, it is not clear what is the output ("solution" in Figure 2) of the model, is it the atom embedding, or the denoised atom coordinate?
>
> Line 82 states: “...representing our model as a denoising function D that provides an estimate of the true coordinates x.” This appears to be fairly clear. See also Equation (1). In addition, in line 86-87 it is stated “and a series of bonded subcomponents whose outputs are summed together for coordinate prediction.”
>
>  * For experiments, it would make results stronger if authors can do comprehensive ablation studies about the proposed novel modules in the model as they are the major novelty contributions.
>
> Each bonded component has an explicit and important purpose, and so, removing any of them would degrade the model. We haven’t performed the experiment to explicitly measure the level of degradation. Early in development, the components were added and tested sequentially.
>
>  * Also, it is recommended to compare with an important baseline method, Torsional Diffusion [1]
>
> Please see our global remarks concerning the paper [1]. In short, we do not consider this work a valid baseline.
>
>  * To make a better organization, it would be better to move Section 3 Datasets to the place just before Section 6 Experiments as they are more closely related.
>
> It is somewhat awkward to talk about the results from training when you haven’t discussed the data set used for training.
>
>  * It is hard to understand what Figure 4 tries to show. What do the y-axis values mean? And how these curves are related to explainability?
>
> Each bonded component is intended to predict a correction to atom positions based on the corresponding physical term (bond length, angle, torsion). To explain how the model determines the total correction to each atom position, we may inspect the output of each bonded component individually. Since each component corresponds to a specific type of physical term, we can understand the output of a component with respect to that term.
>
> Each bonded component uses a MLP to predict a movement distance along a vector. For the bend component, this vector is between the two outer atoms of the bend. The input to the MLP includes the distance between the two outer atoms (the x axis of Figure 4) and the relevant value of the noise schedule (the value of σ, used to distinguish between the various curves presented). The input to the MLP also includes the embedding from the atoms of the molecular structure associated with the bend, implied by the image of the molecule on the left. The vertical line corresponds to the ground truth value established by an independent invocation of the semi-empirical quantum mechanical simulation GFN2-xTB.
>
> Notice that, at the limit of no noise (σ = 0), if the distance between atoms matches the ground truth, this component suggests that no correction is needed. If the distance is too small or too large, a corresponding correction in approximate proportion to the error is applied to bring it to the ground truth. At larger noise values (σ > 0), the curve departs from this behavior, which can be understood as the overcorrection needed for efficient generation at larger noise values.
>
> Figure 4 is provided for one example structure for the bend component. One can consider any interesting molecule structure, with any challenging chemistry (limited only by the imagination), and repeat the process for any component. Or, if a particular molecule performs poorly during generation, one can study that specific molecule, and query specific terms to search for defects.
>
>  * and are recommended to move these results to experiment part to maintain a better organization
>
> Figure 4 pertains to the model and has no direct relationship to the generation process, nor to the experimental results.

---

> > ### Comment · Reviewer_J8rt · 2023-08-13
> > **Follow-up Responses**
> >
> > I appreciate authors' hard work in clarifying their method contributions and addressing my concerns in the rebuttal.
> >
> > - I appreciate authors' clarification and explanation about method details and Figure 4.
> > - For paper organization, I think it is not awkward to move dataset introduction to places after method design. Unless your training method is specially designed for some certain datasets, it is more logically smooth to tell people first what are your method and novelty contributions, then demonstrate the effectiveness of your contributions by experiments on your used datasets (here you may introduce your datasets). It is a common organization style of papers in machine learning community though may not be the habit in other research areas.
> > - I appreciate authors' clarification in the major target of the proposed method in their rebuttal to all reviewers. To my understanding, the major problem to be solved by PIDM is to generating a high-quality initial molecular conformations by creating accurate static terms (bond lengths & angles), and experiments show that PIDM can achieve it in experiments (Table 1). As for variable terms (proper torsions), PIDM does not consider improve their generation accuracy as they may change in different scenarios. I agree that it is not suitable to compare PIDM with Torsional Diffusion (TD). However, I am not persuaded that generating high-quality initial molecular conformations is a really important task in practical applications, as RDKit can also achieve it and produce a good enough initial conformation, this is why TD starts from RDKit produced conformation and refine proper torsions. In my opinion, authors should conduct many more experiments to motivate the importance of generating accurate static terms in producing accurate initial molecular conformations. For instance, in molecular conformation generation problem, authors may show PIDM + TD is much better than original TD (i.e., RDKit + TD); in the problem of generating ligand conformations conditioned on the given ligand and protein pocket [1], authors may show that replacing the commonly used RDKit initialized molecular conformations by PIDM initialized ones leads to much better performance.
> >
> > I suggest to more clearly clarify the central target of PIDM (i.e., focusing on static terms generation) and differentiate PIDM from other molecular conformation generation methods in the revision of paper. Also, more experiments should be added to demonstrate the importance of the studied problem. As these are not easily completed in a short periodic of author-reviewer discussion, I tend to keep my decision of rejection, but encourage authors to make significant reorganization in this paper following my suggestions and resubmit it to a new venue.
> >
> > [1] EQUIBIND: Geometric Deep Learning for Drug Binding Structure Prediction. ICML 2022.

---

### Official Review · Reviewer_htQL · 2023-07-04

**Soundness:** 2 fair
**Presentation:** 1 poor
**Contribution:** 2 fair
**Rating:** 3
**Confidence:** 3

**Summary:**

The paper presents a diffusion model to generate conformers of drug-like molecules. The score model architecture is novel.

**Strengths:**

The authors construct a diffusion model for conformer generation whose score model architecture is inspired by the structure of classical force fields. This is an interesting approach and quite different from the SE(3) or E(3) equivariant graph neural nets that are popular for this kind of problem.

**Weaknesses:**

I found parts of the paper to be quite unclear.

Line 46: I don’t understand this $n\phi - \phi_0$ expression or ‘freedom to select one of $n$ torsion angles’. Is there a discrete set of admissible torsion angles? Sorry, I’m ignorant about molecular geometry – but so will be many of your readers at NeurIPS.

Line 71 ‘attempts to model nonbonded distances which requires it to sample torsional space during training… torsional space is physically ambiguous’: I don’t understand these sentences.

Figure 2: what shape is the output from each of the pink blocks? Is it (number of atoms) * 3? What is the regression target for the summed ‘solution’? In some works on molecule generation or conformation prediction, the regression target is the noise that was added to the atom positions, and in other works the regression target is the ‘clean’ atom positions from the training data, so this needs to be clarified.

Line 125: what is ‘conformer inconsistency at the graph level’?

Section 4 Training: if you want to experiment with different samplers after training then I think that during training you should sample values of $\sigma$ from a continuous range, rather than picking from N discrete values. If you sample by stepping through values of $\sigma$ not seen during training, you will be presenting the score model with inputs it never saw during training, and I expect the performance will be suboptimal.

Figure 4: please explain what the x and y axes are here. In general, I did not understand the ‘explainability’ aspect of the work.

Line 216: I think that the inability of GeoDiff to discriminate between a molecule and its mirror image is a fundamental issue with the GFN architecture.

Table 1 caption says 'best values in each category are highlighted' but I cannot see the highlight (perhaps a printing problem).

The guided generation task in section 7 seems arbitrary. I do not understand equation (12) because in the right-hand-side numerator we have a vector $y$ and then subtract $F(y)$ which appears to be a scalar.

**Questions:**

Line 95 ‘uses bonds as graph edges’:  why not allow some message-passing between non-bonded atoms?

Line 127: is any clustering used to split the data, or can similar molecules appear in both training and test data?

Could you please elaborate on the explainability of the model? For example, show examples of predictions and their explanations. Perhaps this is meant to be shown in Figure 4 but I did not understand it.


**Limitations:**

Yes

---

> ### Author Rebuttal · Authors · 2023-08-08
>
> Thank you for taking the time to read our submission and provide feedback.
>
> Let us begin with your questions.
>
>  * Line 95 ‘uses bonds as graph edges’: why not allow some message-passing between non-bonded atoms?
>
> In chemistry, the classification of chemical groups is associated with bonded connections alone. Thus, we felt there was no need to incorporate message passing for nonbonded atoms.
>
>  * Line 127: is any clustering used to split the data, or can similar molecules appear in both training and test data?
>
> See lines 135-141. Clustering of molecule structures is an interesting problem with no general solution (mapping chemical space to RN is currently an unsolved problem). Instead, we employ simple random selection at the level of molecule structures in order to perform a data split between training, validation, and test subsets.
>
> From experience, we understand that molecular structure data sets tend to have families of closely related compounds. This means that there is the distinct danger that molecules in one part of a random split could be closely associated with molecules in another. To explicitly avoid this issue, we employed a standard structure-matching technique (circular fingerprints and the Tanimoto metric) to discard any molecule in the test data set with a structure that was even remotely similar to any molecule structure in the training or validation sets, or, for that matter, to any molecule in the entire GEOM-drugs data set.
>
>  * Could you please elaborate on the explainability of the model? For example, show examples of predictions and their explanations. Perhaps this is meant to be shown in Figure 4 but I did not understand it.
>
> Each bonded component is intended to predict a correction to atom positions based on the corresponding physical term (bond length, angle, torsion). To explain how the model determines the total correction to each atom position, we may inspect the output of each bonded component individually. Since each component corresponds to a specific type of physical term, we can understand the output of a component with respect to that term.
>
> Each bonded component uses a MLP to predict a movement distance along a vector. For the bend component, this vector is between the two outer atoms of the bend. The input to the MLP includes the distance between the two outer atoms (the x axis of Figure 4) and the relevant value of the noise schedule (the value of σ, used to distinguish between the various curves presented). The input to the MLP also includes the embedding from the atoms of the molecular structure associated with the bend, implied by the image of the molecule on the left. The vertical line corresponds to the ground truth value established by an independent invocation of the semi-empirical quantum mechanical simulation GFN2-xTB.
>
> Notice that, at the limit of no noise (σ = 0), if the distance between atoms matches the ground truth, this component suggests that no correction is needed. If the distance is too small or too large, a corresponding correction in approximate proportion to the error is applied to bring it to the ground truth. At larger noise values (σ > 0), the curve departs from this behavior, which can be understood as the overcorrection needed for efficient generation at larger noise values.
>
> Figure 4 is provided for one example structure for the bend component. One can consider any interesting molecule structure, with any challenging chemistry (limited only by the imagination), and repeat the process for any component. Or, if a particular molecule performs poorly during generation, one can study that specific molecule, and query specific terms to search for defects.
>
> ----
>
> If you have the patience, let us address some of the weaknesses you reported.
>
>  * ‘attempts to model nonbonded distances which requires it to sample torsional space during training… torsional space is physically ambiguous’: I don’t understand these sentences.
>
> Perhaps our global remarks might be useful in this regard.
>
>  * what shape is the output from each of the pink blocks? Is it (number of atoms) * 3?
>
> Correct.
>
>  * In some works on molecule generation or conformation prediction, the regression target is the noise that was added to the atom positions, and in other works the regression target is the ‘clean’ atom positions from the training data, so this needs to be clarified.
>
> The difference between the two is merely semantics, because the difference between truth and noised coordinates is the amount of noise that is added.
>
>  * what is ‘conformer inconsistency at the graph level’?
>
> Some of the conformer structures provided by the authors of the GEOM-drug data set do not match the molecule structure that they are intended to represent. By “graph level”, we mean connectivity (rather than coordinates).
>
>  * if you want to experiment with different samplers after training then I think that during training you should sample values of σ from a continuous range, rather than picking from N discrete values
>
> This option did occur to us. For reasons of time, we never did try this technique, focusing instead on the interesting problem of generation, since this provided the most significant improvement in performance.
>
>  * If you sample by stepping through values of σ not seen during training, you will be presenting the score model with inputs it never saw during training, and I expect the performance will be suboptimal.
>
> We directly investigated the continuity of the output of each bonded component and can confirm that training consistently produced a smooth function of σ.
>
>  * The guided generation task in section 7 seems arbitrary.
>
> It is. This methodology is not fully developed, which is why we describe it as a proof-of-concept.
>
>  * I do not understand equation (12) because in the right-hand-side numerator we have a vector y and then subtract F(y) which appears to be a scalar.
>
> It is hard to see, but F is rendered in bold and is intended to represent a vector.

---

> > ### Comment · Reviewer_htQL · 2023-08-16
> >
> > Thank you for taking the time to respond to the reviews and in particular for clarifying the major target of the paper, which I had not properly understood. I feel that the aim of the paper should be more clearly stated, and you need to do more to persuade the reader that the problem you address is a practically important one that is not adequately solved by existing tools.

---

### Official Review · Reviewer_7Zu8 · 2023-07-06

**Soundness:** 3 good
**Presentation:** 1 poor
**Contribution:** 2 fair
**Rating:** 3
**Confidence:** 5

**Summary:**

A diffusion-based conformation generation method called PIDM is proposed for molecules. It combines several geometries including bond lengths, bend angles, proper torsions, chirality and cis-trans of the noisy conformer as the input, and output the scores of the probability to iteratively generate the conformations. Experiments shows the robustness of the model, while is not convincing enough to show the performance of PIDM.

**Strengths:**

- The proposed model is simple yet effective, with a smaller NFE used compared to GeoDiff.
- Many version of the proposed DIPM has been test in the experiments.

**Weaknesses:**

Several problems exist, so that I think it is not a mature enough article to be published.

1. Confusing presentation and limited novelty.

- First, it is a physics-informed method, but I cannot find supports that there are some physics knowledge as priori knowledge that works in the model. The five geometries are very commonly used that conformation generation methods focus on, such as torsion angle in TorsionalDiff[1] and distance in CGCF[2]. It is not a novel idea based on physics rule. In the guided generation, some of the physics-based or energy-liked terms is used to modified the generating process, but it is not the main contribution of the paper, but a simple experimental attempt.

- Second, the structure of the article is not well-organized. The description of the proposed method is very short, and not completely explained. More details with formulas should be added. The dimension of embedding sizes and other experimental details should be included in the experiment parts. Description of datasets and preprocessing should also placed in the experiment sections. In Sec. 5, most of the details has been given in [3], and the generation process is just an implementation of it, so it can be presented in a brief way, since it is not the main contributions. However, it takes up an entire page, almost the same as the designed method which is your main contributions.
- Third, several parts are not well explained. For example, in Figure 2, what is the red arrow means, and what is the black means. Figure 3 is the loss in the training procedure on training a validation set, but what are you try to tell the readers? If you want to show the robustness of the model, some of evidence about other models are not robust should be compared, since most training/validation loss plots seem to be the same like Figure 3. There are also several similar examples.

2. Symmetries are not considered. As a diffusion models, most methods will consider the SO(3) equivariance or E(3) equivariance of the denoisers, leading the probability model to satisfy $p(x) = p(\Pi x)$ in which $\Pi$ is a matrix representation of the certain group. However, in PIDM, the output is scores of 3D-positions each atoms, but the input is the invariant geometries like bond lengths and angles, so if the noisy positions are rotated, the geometries will be unchanged, and the scores will also not be rotated. The equivariance is not ensured. (I am not very certain about it, because the detailed of model is not fully described in Sec.2, and the input and output of the model is mostly based on my guesses.)

3.  Incomplete experiments.
- The experimental protocols and compared baselines are not convincing. As the new SOTA method, TorsionalDiff shows better experimental performance compared to GeoMol. Both of these two methods generated a conformer based on a prior conformer generated by RDKit. Therefore, at least TorsionalDiff should be included as a baseline, and also, the experimental protocols such as metrics like Recall Coverage/AMR and Precision Coverage/AMR. The GeoDiff is only trained on a very small part of Drugs, but the checkpoint is directly employed for comparison.  Is it reasonable? Or it should be trained on the same training set and test on validation set? For GeoMol as another deep-learning-based models, should it be re-trained and tested with the same protocols to should the superiority of the PIDM?

- The comparison on geometries of generated molecules and ground-truth molecules is interesting, but some detailed comparison should be added. For example, the distribution of C-C, C=C bond distance of baselines, PIDM, and ground truth, or other bend angles, torsion angles, etc., and the JS-diversity between the distributions. Only the MAD cannot fully demonstrate the superiority of the performance.

- In the abstract, it says that the model is resistant to overfitting and explainable. In which part of the experiments gives demonstration of these two advantages of the models. Can PIDM trained on Drugs generalize well on QMugs? If it can, the two datasets consist of molecules of similar sizes (25 ~ 30 atoms),  how about smaller molecules in QM9 or larger ones in ligands in PDBBind? How the model shows explainability?

- The sampling method used is in [3], what is the performance gain it brings compared to the original score-matching sampling methods? If there is no improvements according to empirical comparison, why it is employed? As I know, TorsionalDiff also use a very small NFE during sampling, so in evaluating effectiveness, does PIDM work better?


[1] Bowen Jing, et al, Torsional Diffusion for Molecular Conformer Generation, https://arxiv.org/pdf/2206.01729.pdf

[2] Chence Shi, et al. Learning Gradient Fields for Molecular Conformation Generation, https://arxiv.org/pdf/2105.03902v1)%3C%22

[3] Tero Karras, et al. Elucidating the Design Space of Diffusion-Based Generative Models, https://arxiv.org/pdf/2206.00364.pdf

**Questions:**

The questions are given in the weakness part. Please refer to the points that I am doubtful about.

**Limitations:**

N. A.

---

> ### Author Rebuttal · Authors · 2023-08-08
>
> Thank you for taking the time to review our submission and provide a thoughtful response.
>
> Given the extent of your comments, and the lack of explicit questions, we will attempt to address the least subjective of the weaknesses you have provided.
>
>  * The five geometries are very commonly used that conformation generation methods focus on, such as torsion angle in TorsionalDiff[1] and distance in CGCF[2].
>
> Given that the task of generating conformers is to model physical objects, it is natural to borrow elements from classical force fields. The TorsionalDiff model makes no attempt to model the static conformer terms, relying instead on RDKit, a process that requires no physical insight. Concerning CGCF, to label a bond distance as a physical trait requires little insight, since, after all, it is merely a simple parameter of geometry (the distance between atoms).
>
>  * the generation process is just an implementation of it, so it can be presented in a brief way, since it is not the main contributions.
>
> We disagree. It is remarkable that the training portion of a diffusion model can be separated entirely from generation (i.e., a different number of steps and noising schedules). Presenting multiple options for generation is our way of demonstrating this.
>
>  * Figure 3 is the loss in the training procedure on training a validation set, but what are you try to tell the readers?
>
> We are telling the readers that: (1) during training, losses are uniformly decreasing before reaching a plateau which suggests proper optimization; and (2) the loss from the validation set matches that from training.
>
>  * Symmetries are not considered
>
> Symmetry is implied. The inputs (bond distances, bends, and torsion angles) are all invariant under translation and rotation. The output of each bonded component is a displacement along the vector connecting two atoms, which is invariant under translation and rotation. The loss function is calculated from the distance between model and truth, and is also invariant under translation and rotation.
>
> The symmetry analysis above is obvious using simple geometric principles. We felt no need to emphasize this point.
>
>  * so if the noisy positions are rotated, the geometries will be unchanged
>
> Your concern is misplaced. All Gaussian smearing is uniform (spherical), and so rotations in this respect are irrelevant.
>
>  * As the new SOTA method, TorsionalDiff shows better experimental performance compared to GeoMol
>
> This model is complementary to our model and not comparable. See our global comments for an explanation.
>
>  * the experimental protocols such as metrics like Recall Coverage/AMR and Precision Coverage/AMR
>
> Please see our global remarks concerning the merits of the RMSD metric against a synthetic data set like QMugs. In short, we do not believe that such a metric represents an appropriate protocol for our work.
>
>  * The GeoDiff is only trained on a very small part of Drugs
>
> GeoDiff refers to the CGCF paper for the data set. For that reference, we quote:
>  > “We randomly draw 40,000 molecules and select the 5 most likely conformations for each molecule”
>
> As such, GeoDiff has made the decision to train on a randomly selected 13% of the molecules in the data set and a fraction of the conformers. No explanation is given for why only a portion of the data set is used. Certainly there is no technical barrier. We can assume there was no concern about bias or issue with some sub-population of the data set, since the selection was random. Given no other information, the only conclusion one could draw is that the authors believed that this portion of the data set was adequate for the sake of coverage. We assume the authors stand behind the accuracy of their model so constructed.
>
> Note that we make no such unjustified assumptions and simply run our experiments on the entire data set.
>
>  * For GeoMol as another deep-learning-based models, should it be re-trained and tested with the same protocols
>
> We don’t understand this remark. Certainly the authors of GeoMol are the best judge of the manner in which to train it. Note that these authors, like us, chose to experiment with the entirety of the GEOM-drugs data set.
>
>  * For example, the distribution of C-C, C=C bond distance of baselines
>
> Perhaps you are not familiar with organic chemistry, since the lengths of such bonds are dependent on the chemical groups in which they are found. Some examples of alkane bonds (in Angstroms, predicted by GFN2-xTB) are:
> ```
> 1.52167 propane
> 1.52377 isobutane
> 1.49187 propylene
> 1.49526 acetaldehyde
> 1.49882 isobutylene
> ```
> The MAD of our model indicates a resolution of 0.004 Å, so we would be doing our model a serious disservice by employing such a simplistic measure.
>
>  * or other bend angles, torsion angles, etc.,
>
> In these cases, the chemical group has an even stronger influence. Also, do you have a suggestion on how to deal with bend angles in rings?
>
>  * and the JS-diversity between the distributions
>
> We are puzzled why one would compare distributions when access to individual values are available.
>
>  * Can PIDM trained on Drugs generalize well on QMugs?
>
> See Table 2. In case it was not clear, there is only one test data set (extracted from QMugs) used for this table for all models.
>
>  * If it can, the two datasets consist of molecules of similar sizes (25 ~ 30 atoms), how about smaller molecules in QM9 or larger ones in ligands in PDBBind?
>
> We ignore QM9 (it is an ill conceived data set). As we noted in our submission, there is little point in comparing generated conformers to PDBBind, since structures in the protein data bank, in general, use force field constraints in their experimental solutions and will be biased.
>
> During development, we tested quite large structures one-by-one and inspected the results visually. The model holds up remarkably well. In the supplemental materials we present a generated structure for micafungin, which has 89 heavy atoms, in addition to containing a large macrocycle.

---

### Official Review · Reviewer_4243 · 2023-07-07

**Soundness:** 2 fair
**Presentation:** 2 fair
**Contribution:** 2 fair
**Rating:** 4
**Confidence:** 4

**Summary:**

This paper introduces a physics-informed generative framework targeted at the task of molecular conformation generation. The method is motivated by the formulation of classical force fields and such an idea is reflected via bond, bend, and proper torsion and properly injected into the model design. The experiments are conducted on GEOM-Drugs and QMugs datasets in terms of molecule conformation generation.

**Strengths:**

1. The paper is well-motivated via the lens of classical force fields and the method contains reasonable integration of the idea.

2. The method is easy to follow with necessary details presented to help understand the entire pipeline of the generation framework.

**Weaknesses:**

1. The experimental evaluations lack the necessary details and discussions, which brings concerns about the credibility of the experimental comparison with the baselines (see Q1 and Q3).

2. Important baselines are missing (see Q2).

Minor: Some parts of the presentation of the paper are confusing and may require further checks and polishment, e.g., line 246. In figure 5, is it possible to include some conformations generated by the baselines to offer the readers a clearer qualitative comparison?

**Questions:**

1. In this paper, the authors seem to simply use the open-sourced checkpoints of the baselines like GeoMol and GeoDiff to report the testing results. However, it is uncertain whether the entire training protocol is fair compared with the original setups in these baseline papers. More justifications should be addressed on this point to ensure that the experiments offer a fair comparison.

2. Important baselines, e.g., Torsional Diffusion [1], are missing and should be discussed and compared since they share a similar/relevant idea of leveraging physical priors of bond/bend/torsional angles.

3. Why do the evaluation protocol and metrics seem so different from previous works like GeoMol and GeoDiff? Is it possible to conduct experiments following the widely-adopted setup in order to make a convincing comparison?


[1] Jing, Bowen, et al. "Torsional diffusion for molecular conformer generation." Advances in Neural Information Processing Systems 2022.

**Limitations:**

The authors have satisfactorily discussed the limitations.

---

> ### Author Rebuttal · Authors · 2023-08-08
>
> We want to thank the reviewer for reading our submission and providing valuable feedback.
>
> We would like to begin by answering the given questions.
>
> (1) Using open-sourced checkpoints
>
> The goal of a conformer generator is to predict conformers in a manner that respects the underlying physics of a molecule. That physics is independent of how a model was trained, assuming, of course, that the training set consists of physically valid samples. As long as the training set has sufficient coverage, it is also independent of the molecule structures contained in that training set.
>
> What is wonderfully convenient is that both the QMugs and GEOM-drugs data sets use the same physical basis for their contents: the semi-empirical method GFN2-xTB. We are assuming that authors of both QMugs and GEOM-drugs used this method correctly. Therefore, the physical characteristics of the conformers found in both sets are directly comparable.
>
> The remaining question is whether the chemical space of the two data sets sufficiently overlap. We will remind the reviewer that both data sets are intended to represent a diverse selection of drug-like molecules. We will also contest that the atom types contained within the QMugs data set is a subset of the atom types found in the GEOM-drugs data set. Therefore, any model trained on (the full or a random subset of) the GEOM-drugs data set should be expected to reproduce the physical characteristics of (at least the vast majority of) the conformers in the QMugs data set.
>
> You may have noticed that we applied this principle to our own models. That is, our models trained on the GEOM-drug data set were compared to the test data set derived from the QMugs data set. You may have also noticed that performance was not significantly affected.
>
> In addition, it is our contention that a model that proposes to reproduce conformers for drug-like molecules should be expected to perform correctly on drug-like molecules outside of its training set. We will admit, though, that what it means to be “drug-like” can be a contentious topic.
>
> As you may have observed, we went to some trouble to ensure that the test data set was composed of molecule structures that did not structurally overlap with the GEOM-drugs data set in its entirety. This was to avoid data leakage. It also ensures that any model trained on any subset of the GEOM-drugs data set did not use any molecule in the test data set in its training.
>
> (2) Important baselines
>
> Please see our global remarks concerning the paper [1]. In short, we do not consider this work a valid baseline.
>
> (3) Evaluation protocol
>
> Please see our global remarks concerning the merits of the RMSD metric against a synthetic data set like QMugs. In short, we do not believe that such a metric represents an appropriate protocol for our work.
>
> ---
>
> If you will bear with us, we would like to take some additional space to address the weaknesses you have listed.
>
> We apologize for the lack of details and discussion. This, unfortunately, was due to the very tight space requirements. The complete details of the model are provided in the supplemental materials. If you have suggestions on how we can improve our submission, within the space limitations, then we welcome any feedback. In particular, we are interested if you believe that material was extraneous or duplicated, or if that major sections, such as on directed generation, deserve to be sacrificed in order to accommodate more model details.
>
> Concerning figure 5, we made the judgment that providing additional examples from our work was more important than providing an example from one of the baselines. If we had to select a baseline, we would likely select RDKit. Would this be an addition that you believe would improve the presentation? Please note that for the sake of honesty, we did not cherry pick these examples, but chose them at random. Thus, interesting examples were not guaranteed. We did provide hand-selected examples of interest in the supplemental materials, along with many other (perhaps too many) randomly selected samples.

---

### Author Rebuttal · Authors · 2023-08-08

We think it is important to clarify certain aspects of molecule conformers, pertaining, in particular, to their use in drug discovery. As domain experts on this particular topic, we can speak authoritatively, and encourage the reviewers to reach out to other domain experts if they have further questions.

As discussed in our paper, one can consider two types of degrees of freedom when establishing the conformer of a molecule:

 1) Static terms. This includes bond lengths, bond angles, and improper torsions. Bonded interactions generally describe geometric configurations that are (approximately) fixed for all valid conformers. For example, within a set of valid conformers for a molecule, you will find that the bond length between two given atoms are all the same.
 2) Variable terms. These are the proper torsions. They have the freedom to vary and can be used to distinguish between the individual members of a set of conformers of a given molecule.

Given that varying the proper torsions allows you to move between molecule conformers, it is the proper torsions that are typically varied during ligand-protein docking. This is how Autodock, Gold, and Glide all function. Note that adjusting the proper torsions only allows one to convert between conformers. One must start with a valid conformer consisting of valid static terms.

The QMugs and GEOM-drugs data sets are based on conformers generated in vacuum. It should be emphasized that this choice is highly unnatural for drug-like molecules because it corresponds, physically, to a dilute gas. Drug-like organic molecules are almost always found either in solution or in solid form. Or, in the case of PDB structures, as bound to a protein. All of these physically realistic environments produce dramatically different energies than vacuum. To put it succinctly, the set or proper torsion angles found in these data sets have practically no physical significance.

Our solution to this data set dilemma is to not be overly concerned about which of the favored values of proper torsion angle are selected during generation. This could be criticized as lazy. However, we will point out that for many applications, such as ligand-protein docking, the choice of proper torsion angle is irrelevant, because these angles are subsequently manipulated by the docking algorithm. In addition, it is a minor algorithmic task (with the exception of macrocycles) to sample torsion angles in any required postprocessing, in order to select that set of torsion angles which are most applicable to the task at hand (for example, the angles that produce the lowest potential energy for a given environment).

This brings us to the subject of the RMSD. For the static terms, since the associated atoms are in close proximity, errors in quantities like bond length and angle produce little difference in RMSD. An incorrect bond angle can introduce a large change in a distant atom, because of leverage, but in most molecules, it is possible to compensate by altering a proper torsion angle. As such, the RMSD of the entire molecule is a poor way of measuring accuracy of static terms of a molecule.

In contrast, altering the proper torsion angle of a molecule can introduce dramatic differences in RMSD. As such, the RMSD metric, when applied to the molecule as a whole, is primarily a measure of the choice of proper torsion angles. Unless we are interested in reproducing the selection of favored proper torsion angles contained in the test data set, the RMSD metric is, for the most part, useless as a benchmark.

We understand that some research efforts in machine learning have relied on the RMSD benchmark against synthetic data sets in their work. Nevertheless, the popularity of a benchmark, in our opinion, is not an appropriate reason to dismiss the drawbacks discussed above.

This brings us to the question of selecting a good, physically meaningful benchmark. If we are interested in establishing the accuracy of reproducing the static terms of molecule conformers, then one choice is to directly compare those terms between generated conformers and an independent, test data set. This was our choice.

Let’s discuss the paper [1] mentioned by some of the reviewers. We contend that this work is not a suitable baseline for our work. The reason is simple: the goal of this model is opposite and complementary to ours. Here is quote taken directly from the paper:

> “We instead propose torsional diffusion, in which the diffusion process over conformers acts only on the torsion angles and leaves the other degrees of freedom fixed. This is possible and effective because the flexibility of a molecule, and thus the difficulty of conformer generation, lies largely in torsional degrees of freedom [Axelrod and Gómez-Bombarelli, 2022]; in particular, bond lengths and angles can already be determined quickly and accurately by standard cheminformatics methods.”

Thus, the authors have constructed a model with the sole purpose of sampling from the set of favored proper torsion angles, a task that we have deliberately chosen not to address. They rely entirely on RDKit for the challenging task of establishing the static terms of conformers. Given that our choice of benchmark scheme is independent of the selection from the set of favored proper torsion angles, including [1] directly serves no purpose, since we would only be measuring RDKit a second time.

[1] Jing, Bowen, et al. "Torsional diffusion for molecular conformer generation." Advances in Neural Information Processing Systems 2022.

---

### Decision · Program_Chairs · 2023-09-21

**Decision:**

Reject

**Comment:**

This paper presents a novel generative model called PIDM for generating 3D molecular conformations from 2D molecular graphs. The proposed approach utilizes an ODE-based diffusion model to generate 3D molecular conformations and designs a noise removal network consisting of multiple modules to capture various physical information from the molecule. The proposed method outperforms two baseline methods, GeoMol and GeoDiff, in experiments. While it presents an interesting approach, a significant issue with this paper lies in the insufficient explanation of many necessary details of the proposed method.
The central focus of PIDM should be clarified further, and its superiority over other molecular structure generation methods with PIDM needs more substantial support. Based on valuable discussion during the rebuttal period and feedback from reviewers, it is recommended to extensively reorganize the paper and consider resubmission to a new venue.